# The Evaluation of Radiolabeled Prostate-Specific Membrane Antigen Positron Emission Tomography/Computed Tomography for Initial Staging in Intermediate-Risk Prostate Cancer Patients: A Retrospective Multicenter Analysis

**DOI:** 10.3390/diagnostics14232751

**Published:** 2024-12-06

**Authors:** Laura Evangelista, Priscilla Guglielmo, Giulia Giacoppo, Lucia Setti, Demetrio Aricò, Lorenzo Muraglia, Katia Marzo, Nicolò Buffi, Vittorio Fasulo, Marcello Rodari, Jelena Jandric, Antonio Salvaggio, Manuela Bonacina, Massimo Lazzeri, Giovanni Lughezzani

**Affiliations:** 1Nuclear Medicine Unit, IRCCS Humanitas Research Hospital, 20089 Rozzano, Italy; lorenzo.muraglia@humanitas.it (L.M.); katia.marzo@humanitas.it (K.M.); marcello.rodari@humanitas.it (M.R.); jelena.jandric@humanitas.it (J.J.); 2Department of Biomedical Sciences, Humanitas University, 20072 Pieve Emanuele, Italy; nicolo.buffi@hunimed.eu (N.B.); vittorio.fasulo@hunimed.eu (V.F.); giovanni.lughezzani@hunimed.eu (G.L.); 3Nuclear Medicine Unit, Humanitas Gavazzeni, 24125 Bergamo, Italy; priscilla.guglielmo@gavazzeni.it (P.G.);; 4Nuclear Medicine Unit, Humanitas Istituto Clinico Catanese, 95045 Misterbianco, Italy; giulia.giacoppo@humanitascatania.it (G.G.); demetrio.arico@humanitascatania.it (D.A.); 5Urology Unit, IRCCS Humanitas Research Hospital, 20089 Rozzano, Italy; massimo.lazzeri@humanitas.it; 6Urology Unit, Humanitas Mater Domini, 21100 Castellanza, Italy; 7Urology Unit, Humanitas Istituto Clinico Catanese, 95045 Catania, Italy; antonio.salvaggio@humanitascatania.it

**Keywords:** prostate cancer, intermediate risk, PSMA PET/CT, diagnostic performance, change in management

## Abstract

Objectives. The aim of the present study was to assess the performance of radiolabeled-PSMA PET/CT in a cohort of intermediate-risk prostate cancer (PCa) patients for initial staging. Methods. This is a retrospective, multicenter analysis of patients diagnosed with intermediate-risk PCa who were staged using radiolabeled PSMA PET/CT to evaluate the extent of the disease before initiating appropriate treatment. The study included patients from the Nuclear Medicine Units of the Humanitas group between 2021 and 2024. The change in management due to the PSMA PET/CT examination was assessed. Results. A total of 181 patients were enrolled across all three centers. Histopathological assessment from biopsy revealed that 51.4% of patients had favorable PCa, while 48.6% had unfavorable disease. PET/CT was positive for the primary lesions in all patients, but it revealed a positivity rate in 23 (12.7%) patients for nodes and distant organs, with a positivity rate of 0.21 in the unfavorable group and 0.05 in the favorable group (*p* < 0.005). Based on follow-up data, diagnostic accuracy was higher than 90% in both the favorable and unfavorable groups for lymph node and distant metastases. The inclusion of PSMA PET/CT in the diagnostic algorithm for patients with intermediate-risk PCa impacted patient management in 24 (13.3%) cases, based on the multidisciplinary team decision. Conclusions. PSMA PET/CT can affect the management of patients with intermediate-risk PCa in up to 13% of cases, mainly for unfavorable diseases. New imaging techniques as a first-line imaging procedure can help to plan the correct therapeutic approach in the intermediate-risk PCa group.

## 1. Introduction

Prostate cancer (PCa) represents the most diagnosed cancer in men and the second most common cause of cancer death in Western countries [1]. While in most cases localized PCa can be potentially eradicated with radical prostatectomy or definitive radiotherapy, when the disease has already spread to the lymph nodes or to other sites, local treatments are not enough [1,2,3]. Therefore, identifying clinical lymph nodes and metastatic-positive patients is mandatory in order to choose the best therapeutic strategy. Several imaging technologies have been used for PCa staging purposes [4,5,6,7,8,9,10,11,12,13]. Conventional staging, based on abdominopelvic computed tomography (CT) and bone scans, has often been shown to have insufficient sensitivity, especially when staging men with high-risk PCa [5,6,7,8,9,10]. The limited performance of conventional CT and bone scanning in detecting nonlocalized PCa during primary staging can lead to inadequate treatment decisions. Due to the limitations of conventional imaging, several preoperative tools, including the Briganti, Partin, and Memorial Sloan Kettering Cancer Center (MSKCC) nomograms, as well as the Roach formula, have been developed to better predict the individual risk of lymph node metastasis [1]. However, since these tools have also demonstrated suboptimal accuracy, pelvic lymph node dissection (LND) is currently considered the most reliable approach to assessing nodal involvement [1].

The field of non-invasive nodal and metastatic staging of PCa is evolving very rapidly. New imaging technologies, such as radiolabeled prostate-specific membrane antigen positron emission tomography with computed tomography (PSMA PET/CT) and multiparametric magnetic resonance imaging (mpMRI), provide a more sensitive detection of LN and bone metastases than the classical work-up with bone scanning and abdominopelvic CT [11,12,13]. PET/CT with [68Ga]-[18F]-PSMA agents have emerged as a highly sensitive and specific modality for the initial staging of PCa, mainly in patients at high and very high risk for the disease [11,13]. Therefore, there is growing evidence supporting the use of PSMA PET/CT for primary staging in the replacement of conventional imaging. However, there are still some open questions about the use of this technology in the specific risk category of patients, such as patients with intermediate-risk disease [14,15]; indeed, most published studies are relative to mixed populations (i.e., intermediate and high risks). Furthermore, the emerging use of specific scores from PSMA PET/CT images, such as primary score, PSMA-RADS, and others [16,17], for subclassifying recurrence risk and histopathological patterns (favorable vs. unfavorable) [18], complicates the initial evaluation of these patients.

The aim of the present study was to assess the performance of radiolabeled PSMA PET/CT in a cohort of intermediate-risk PCa patients for initial staging.

## 2. Materials and Methods

### 2.1. Study Design and Patient Selection

This is a retrospective, multicenter analysis of patients diagnosed with intermediate-risk PCa (both favorable and unfavorable, following the guidelines by Mohler et al. [19] and Sanda et al. [20]). These patients were staged using radiolabeled PSMA PET/CT to evaluate the extent of the disease before initiating appropriate treatment. The study included patients from the Nuclear Medicine Units of the Humanitas group (Cancer Center in Rozzano, Milan, Italy; Gavazzeni in Bergamo, Italy; and Istituto Clinico Catanese in Catania, Italy) between 2021 and 2024. Inclusion criteria were as follows: (1) age > 18 years; (2) patients with a well-established diagnosis of intermediate-risk PCa (ISUP grade: 2 or 3); and (3) patients who underwent 68Ga-PSMA-11 or 18F-PSMA-1007 PET/CT. Exclusion criteria were as follows: (1) previous history of other oncological diseases; (2) previous radiation treatments, androgen-based therapies, or chemotherapies; and (3) missing or unavailable clinical and follow-up data. This study was conducted in accordance with the principles of the Declaration of Helsinki (1964) and received approval from the Ethical Committee (n.169/24, date: 19 March 2024).

### 2.2. PSMA PET/CT Protocol

All patients underwent PSMA-ligand PET/CT scan with either [68Ga]Ga-PSMA-11 or [18F]F-PSMA1007. The 18F-PSMA-1007 precursor, cassettes, and reagents for the synthesis were acquired from ABX Advanced Biochemical Compounds (Radeberg, Germany). 18F-PSMA-1007 was synthesized in a TRASIS AllInOne Module following the prescribed step-by-step procedure. The entire synthesis process, from start to transfer, takes approximately 45 min, yielding a final product volume of 20 ± 1.0 mL. The average synthesis yield of 18F-PSMA-1007 is approximately ≥40% ± 10%, without correction for decay. The radiochemical purity of the final product exceeds 95%. The MiniAllInOne module was used to synthesize 68Ga-PSMA-11. The precursor, cassettes, and reagents for this synthesis were acquired from Trasis (Ans, Belgium). The entire synthesis process, from start to transfer, takes approximately 20 min, yielding a final product volume of 10 ± 0.5 mL. The average synthesis yield of 68Ga-PSMA-11 is approximately ≥65%, without correction for decay. The radiochemical purity of the final product exceeds 98%, as estimated by high-performance liquid chromatography. Appendix A reports the information about tracer production at each center.

PSMA-ligand PET/CT was performed according to versions 1.0 and 2.0 of the European Association of Nuclear Medicine (EANM) guidelines for prostate cancer imaging [21]. PET/CT images were acquired between 60 and 90 and between 90 and 120 min after [68Ga]Ga-PSMA-11 or [18F]F-PSMA-1007, respectively. All patients underwent oral hydration and bladder voiding prior to PET/CT imaging. In selected cases, furosemide was administered 30 min after PSMA-ligand injection (20 mg intravenously). PET/CT images were acquired using different PET/CT scanners (Appendix A). Co-registered CT data were used for the attenuation correction of PET images.

### 2.3. Image Analysis

All images were reviewed by two nuclear medicine physicians with at least 5 years of experience with PET using dedicated software PET/CT scans. PET/CT images were defined as negative if no area of increased radiopharmaceutical uptake was observed compared to the background. The criterion for positivity was at least one abnormal area of radiopharmaceutical uptake outside the physiological distribution or higher than the surrounding physiological activity, respectively, for 68GaPSMA-11 and 18F-PSMA-1007. Both visual and qualitative analyses were used. Semiquantitative analysis in terms of maximum standardized uptake value (SUVmax) was calculated by using an isocontour voxel of interest (VOI). Moreover, in the primary prostate lesions, the primary score was documented as previously defined, with 5 categories: score 1, no significant pattern within the prostate; score 2, a diffuse transition or central zone pattern; score 3, focal transition zone activity above twice the background transition zone counts; score 4, focal peripheral zone activity of any intensity; and score 5, an SUV of more than 12 [16].

All imaging studies were classified in terms of diagnostic accuracy for lymph node (N) and distant metastasis (M) using per-patient-based analysis, by using the following criteria: true positive (TP), patients with a positive PSMA PET, and the evidence of disease on histopathology in the case of radical prostatectomy ± LAD, or follow-up PSA trend or any imaging after treatments; true negative (TN), patients with a negative PSMA PET/CT and no evidence of disease at histopathology or follow-up; false positive (FP), patients without any evidence of disease on histopathology/follow-up, but a positive PSMA PET/CT; false negative (FN), patients with evidence of disease on histopathology/follow-up but a negative PSMA PET/CT.

### 2.4. Multiparametric MRI Imaging

Reports from mpMRI examinations for each patient were evaluated, and if available, the images were reassessed by a dedicated radiologist. Prostate Imaging Reporting and Data System (PI-RADS) Version 2.1 [22] was used as the reference to score each lesion.

### 2.5. Change in Management

The change in management was defined as a change of therapeutic strategies (i.e., from the local to systemic treatment, defined as major) or a change of therapeutic approach (i.e., radiation planning, tailoring of LND template based on PSMA PET/CT results, or a combination of local and/or systemic treatments, defined as minor).

### 2.6. Statistical Analysis

The primary purpose of the present study was to assess the effect of radiolabeled PSMA PET/CT on patient management in a cohort of intermediate-risk PCa patients referring to the Humanitas group’s hospitals for initial staging. The secondary endpoints were (1) to determine the additional diagnostic value provided by the primary score from PSMA PET/CT in this cohort of patients and (2) to compare the detection of disease provided by radiolabeled PSMA PET/CT with that by mpMRI, in a subgroup of patients.

A sample size of approximately 200 patients was determined to achieve a power of over 80% to detect a 10–15% change in therapeutic approach due to the inclusion of PSMA PET/CT in the diagnostic algorithm. The distribution of data was assessed by using the Shapiro–Wilk test. Descriptive statistics regarding patient, imaging, and tumor characteristics as well as preoperative and postoperative data were provided for the whole population. Medians and ranges were reported for normally and non-normally distributed continuous variables. Categorical variables were presented as frequencies and proportions. Statistically significant differences between groups were assessed using Mann–Whitney and Wilcoxon rank-sum tests for non-normally distributed noncontinuous variables and Pearson chi-square tests for categorical variables. Sensitivity, specificity, positive predictive value, and negative predictive value of PSMA PET/CT for N and M staging purposes were determined and their 95% confidence intervals were calculated. Concordance between different imaging tools (PSMA PET/CT and MRI) was measured using Cohen’s kappa coefficient. The significance was set at 0.05. All analyses were performed with MedCalc^®^.

## 3. Results

A total of 181 patients were enrolled across all three centers. The characteristics of the study population are reported in Table 1. Histopathological assessment from biopsy revealed that 51.4% of patients had favorable PCa, while 48.6% had unfavorable disease. The PSA levels were significantly higher in the unfavorable group compared to the favorable group (*p* < 0.05), and treatment types also differed, with local treatment being more common in the favorable group than the unfavorable group (94.7% vs. 69.3%, *p* < 0.05). All patients underwent PET/CT before any treatment approach, with 78 (43.1%) receiving a [68Ga]Ga-PSMA-11 scan and 103 (56.9%) undergoing an [18F]F-PSMA-1007 examination. PET/CT was positive for the primary lesions in all patients, but it revealed positivity in 23 (12.7%) patients for nodes and distant organs. In particular, PSMA PET/CT identified lymph node and distant metastases with a higher positivity rate in the unfavorable group than in the favorable group (14.8% vs. 2.2%, *p* < 0.005, and 13.6% vs. 5.4%, *p* = 0.0576, respectively). The primary tumor was monofocal in 124 (68.5%) patients, occurring more often in favorable than in unfavorable patients (34.4% vs. 28.4%), although this difference was not statistically significant (*p* = 0.386). The primary scores were distributed as follows: score 1 in 17 (9.4%), score 2A and 2B in 19 (10.5%), score 3 in 16 (8.8%), score 4 in 75 (41.4%), and score 5 in 54 (29.8%). The rate of a primary score ≥4 was higher in the unfavorable group compared to the favorable group (*n* = 74; 84.1% vs. *n* = 75; 59.1%, *p* < 0.005). PSMA positivity at the lymph node level was found in patients with a primary score ≥ 4 both in favorable and unfavorable risk groups. Conversely, distant metastasis positivity at PSMA PET/CT was found in 80% and 92% of patients with primary scores ≥ 4, respectively, for favorable and unfavorable risk groups. Based on follow-up data (histopathology was available in 105 patients), diagnostic performance was high in both the favorable and unfavorable groups, for both lymph node and distant metastases (Table 2). The positivity rates in lymph node and distant metastasis for [18F]F-PSMA-1007 vs. [68Ga]Ga-PSMA-11 PET/CT were 8/103 (7.8%) vs. 7/78 (8.9%) and 6/103 (5.8%) vs. 11/78 (14.1%), respectively. However, in patients with the availability of the standard of reference, false positive findings were equally distributed between [18F]F-PSMA-1007 and [68Ga]Ga-PSMA-11 PET/CT, either for lymph node or distant metastases.

### 3.1. MpMRI and PSMA PET/CT

Both MRI and PSMA PET/CT images were available for 136 (75%) patients. At MRI, the PIRADS 2.1 scores were distributed as follows: score 1 in 3 (2.2%) patients, score 2 in 6 (4.4%), score 3 in 10 (7.4%), score 4 in 84 (61.8%), and score 5 in 33 (24.3%) patients. The primary tumor was unifocal in 100 (79.4%) patients according to MRI, showing poor agreement with PSMA PET/CT (K-Cohen = 0.14; CI95%: 0.03–0.31). Specifically, 29/100 (29%) patients identified as having monofocal tumors upon MRI were considered to have plurifocal tumors on PSMA PET/CT. MRI detected capsular invasion in 10 (7.4%) cases and seminal vesicle invasion in 6 (4.4%) cases. The rates of capsular and seminal vesicle invasion were higher in the unfavorable group compared to the favorable group (10.6% vs. 4.3% for capsular invasion and 7.6% vs. 1.4% for seminal vesicle invasion), although these differences were not statistically significant. Notably, a PIRADS 2.1 score of ≥4 was slightly more frequent in the unfavorable group than in the favorable group (87.9% vs. 84.2%), but this was also not statistically significant. Figure 1 illustrates the distribution of PIRADS 2.1 and primary scores in the favorable and unfavorable patient groups and their correlation with PSMA PET/CT findings.

No findings suggestive of lymph node or distant metastases were reported upon MRI for any of the 136 patients. Neither demographic nor MRI findings were significantly associated with a positive PSMA PET/CT in either the favorable or unfavorable risk categories. There was a slightly increased number of patients at unfavorable risk showing vesicle and capsular invasion (Table 3).

### 3.2. PSMA PET/CT and Change Management

The inclusion of PSMA PET/CT in the diagnostic algorithm for patients with intermediate-risk PCa impacted patient management in 24 (13.3%) cases, based on the multidisciplinary team decision. The impact was minor in 10 (41.7%) cases and major in 14 (58.3%) cases. Notably, PSMA PET/CT affected the management of patients with both favorable and unfavorable diseases in similar proportions (12% vs. 15%, respectively). However, major changes were more frequently observed in the unfavorable risk group compared to the favorable risk group (*n* = 8, 61.5% vs. *n* = 2, 18.2%; *p* < 0.05; Figure 2). Additional information about the change in management before and after PSMA PET/CT is reported in Table 4.

## 4. Discussion

The present study found that PSMA PET/CT had a positivity rate for nodal and distant organs for 12% of patients with intermediate-risk PCa, with a higher rate in the unfavorable risk group compared to the favorable risk group (21% vs. 5%). The inclusion of PSMA PET/CT in the diagnostic workflow affected therapeutic management in 13% of cases, with a major impact on 62% of these patients.

This underscores the utility of PSMA PET/CT in intermediate-risk unfavorable PCa, suggesting an additional indication for this advanced imaging modality. Conversely, the effect of PSMA PET/CT in favorable intermediate-risk PCa patients was limited in terms of positivity rate and management changes, with major changes in about 18% of cases. Dekalo et al. [23] reported a PSMA PET/CT positivity rate for lymph node involvement of 5% in patients with favorable intermediate-risk PCa, with a sensitivity of 50% and a PPV of 25% based on histopathological analysis. The authors concluded against using PSMA PET/CT in this cohort due to its low clinical relevance compared to current practices such as nomograms. Conversely, Hagens et al. [15], in a study of 396 patients with newly diagnosed unfavorable intermediate-risk PCa, found a PSMA PET positivity rate of 9.3% for both lymph node and distant metastases but did not report diagnostic performance data. Chikatamarla et al. [24] found higher incidence rates for metastatic disease using [18F]F-PSMA-1007 PET/CT for primary PCa staging. Within their intermediate-risk PCa population, 8.5% (6/71) of men had bone metastases on PSMA PET/CT, which could be attributed to [18F]F-PSMA-1007’s higher rates of nonspecific bone lesions [25,26,27]. Similarly, we found that 8/181 (4.4%) patients had false positive results on PSMA PET/CT, mainly for bone metastases, with similar frequencies for both 68Ga and 18F PSMA agents. The interpretation of the imaging should consider both clinical and experience data; in the present study, the images were carefully reinterpreted by nuclear medicine physicians with more than 10 years of experience in PCa imaging. The PIRADS score is widely used for defining PCa risk at initial diagnosis, but the primary score has also been described [16]. In this study, the primary score was used to assess the prediction of distant metastases in patients with favorable and unfavorable diseases. Patients with unfavorable diseases had both a higher positivity rate and a higher primary score. In contrast, no difference was observed in PIRADS scores, suggesting the potential utility of the primary score as a “biomarker” for predicting widespread disease in intermediate-risk PCa.

Current EAU guidelines recommend that initial staging for intermediate-risk patients include at least cross-sectional abdominopelvic CT imaging and a bone scan for metastatic screening, with PSMA PET/CT being used if available to increase accuracy [28]. However, these guidelines do not differentiate between favorable and unfavorable risk, and the evidence is considered weak. NCCN guidelines suggest using conventional imaging or new imaging technologies such as PSMA PET/CT only in patients with unfavorable PCa risk, while advocating for active surveillance in favorable PCa risk patients [29]. Our study found a change in management in up to 13% of patients with intermediate-risk PCa, highlighting the importance of accurate staging, as intermediate-risk is the most prevalent, accounting for over 40% of PCa cases.

Our study has some limitations. First, given the retrospective nature of this study, the presence of selection bias cannot be ruled out, meaning results should be interpreted with caution. Second, there was a lack of histopathologic evaluation in all patients, because the majority of patients with a PSMA PET/CT positivity were treated with radiotherapy (with/without a hormonal agent), and no re-evaluation by a central anatomical pathologist for the biopsy results was made. Furthermore, although PSMA PET/CT scans were performed in three hospitals, the scan protocols were standardized to limit their influence on the final findings. Additionally, two different PSMA agents were used; however, the PSMA images were reassessed by a reader with solid experience in PCa imaging. Moreover, only a per-patient-based analysis was performed. Finally, the number of patients needed to have statistical power was not reached (181 vs. 200), which makes the article, at best, hypothesis-generating.

## 5. Conclusions

In conclusion, PSMA PET/CT can affect the patient management of patients with intermediate-risk PCa in up to 13% of cases, mainly in cases of unfavorable diseases. The present study underlined the need to opt for new imaging techniques as a first-line imaging procedure to plan the correct therapeutic approach on a large scale. Future prospective studies are needed to further investigate the diagnostic value of PSMA PET/CT in this highly incident PCa patient population. Moreover, the inclusion of PSMA PET/CT, as a diagnostic imaging modality, is strongly recommended in future clinical trials.

## Figures and Tables

**Figure 1 diagnostics-14-02751-f001:**
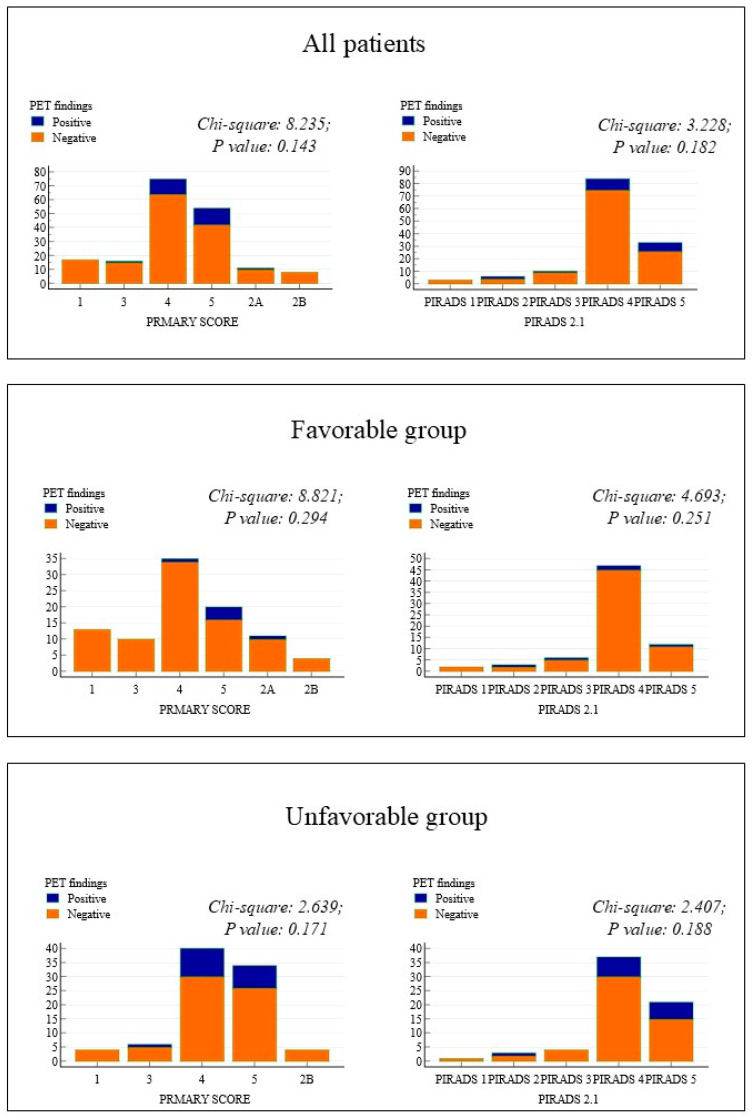
Distribution of PIRADS 2.1 and primary score in accordance with PSMA PET/CT results (positive vs. negative for lymph node and distant metastases).

**Figure 2 diagnostics-14-02751-f002:**
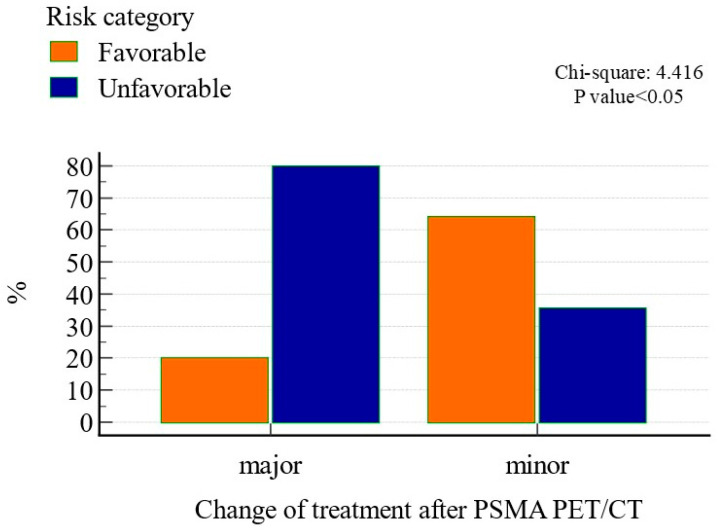
Impact of PSMA PET/CT on patient management based on the intermediate-risk group (favorable vs. unfavorable).

**Table 1 diagnostics-14-02751-t001:** Characteristics of the study population.

Variables		Risk Category	
		Favorable	Unfavorable
N of patients	181	93	88
Median (range) age in years	70 (48–84)	69 (52–84) *	72 (48–83) *
Initial PSA			
Median (range), ng/mL	6.6 (1.17–20)	6.1 (1.9–12.9) *	7.2 (1.2–20) *
ISUP (biopsy), *n* (%)			
Favorable (2)	93 (51.4%)		-
Unfavorable (3)	88 (48.6%)	-	
Type of treatment			
ADT + new-generation HT	1 (0.5%)	0 *	1 (1.1%) *
Surgery	108 (59.7%)	66 (71%) *	42 (47.7%) *
HIFU	6 (3.3%)	5 (5.4%) *	1 (1.1%) *
ADT	6 (3.3%)	0 *	6 (6.8%) *
RT	35 (19.3%)	17 (18.3%) *	18 (20.6%) *
RT + ADT	24 (13.4%)	4 (4.3%) *	20 (22.7%) *
TURP + ADT	1 (0.5%)	1 (1.1%) *	0 *

ADT = androgen deprivation therapy; HIFU: high-intensity focused ultrasound; RT = radiotherapy; TURP: transurethral resection of the prostate; * *p* < 0.05.

**Table 2 diagnostics-14-02751-t002:** Diagnostic performance of PSMA PET/CT for N and M in favorable and unfavorable groups.

	TP	TN	FP	FN	Sens (95%CI)	Spec (95%CI)	PPV (95%CI)	NPV (95%CI)	Acc (95%CI)
N *–All	13	166	2	0	100	99 (97–100)	87 (68–100)	100	99 (97–100)
N–Fav	1	91	1	0	100	99 (97–100)	50 (1–100)	100	99 (97–100)
N–Unf	11	76	1	0	100	99 (96–100)	91 (75–100)	100	99 (97–100)
M *–All	9	164	8	0	100	95 (92–98)	53 (20–86)	100	96 (92–98)
M–Fav	2	88	3	0	100	97 (93–100)	40 (27–100)	100	97 (93–100)
M–Unf	7	76	5	0	100	93 (88–99)	58 (22–94)	100	94 (89–99)

* For the lymph node assessment, histopathological analysis was available in 105 patients, while follow-up was used in the residual 76 patients. All = all patients (*n* = 181); Fav = favorable (*n* = 93); Unf = unfavorable (*n* = 88); TP = true positive; TN = true negative; FP = false positive; FN = false negative; Sens = sensitivity; Spec = specificity; PPV = positive predictive value; NPV = negative predictive value; Acc = accuracy.

**Table 3 diagnostics-14-02751-t003:** Correlations among demographics, MRI findings, and PSMA PET/CT results.

Variables	Risk Category					
	Favorable			Unfavorable		
	Negative PET/CT	Positive PET/CT	*p* Value	Negative PET	Positive PET	*p* Value
N of patients	87	6		69	19	
Median (range) age in years	69 (52–84)	74 (58–78)	0.219	71 (48–83)	73 (58–83)	0.239
Initial PSA Median (range), ng/mL	6.2 (1.9–12.9)	5.1 (4.2–10)	0.578	7 (1.2–20)	8 (2.9–20)	0.156
N patients available for MRI	65	5		52	14	
MRI						
Monofocal	49 (75%)	4 (80%)	0.153	37 (71%)	10 (71%)	0.976
Plurifocal	11 (17%)	0		12 (23%)	3 (21%)	
Missing	5 (8%)	1 (20%)		3 (6%)	1 (7%)	
MRI–PIRADS 2.1						
1	2 (3%)	0		1 (2%)	0	
2	2 (3%)	1 (20%)	0.251	2 (4%)	1 (7%)	0.661
3	5 (8%)	1 (20%)		4 (8%)	0	
4	45 (69%)	2 (40%)		30 (58%)	7 (50%)	
5	11 (17%)	1 (20%)		15 (28%)	6 (43%)	
MRI EPE						
No	61 (94%)	3 (60%)	<0.005	46 (88%)	10 (72%)	
Yes	1 (2%)	2 (40%)		4 (8%)	3 (21%)	0.308
Doubtful	3 (4%)	0		2 (4%)	1 (7%)	
MRI SVI						
No	64 (98%)	5 (100%)	0.033	49 (94%)	12 (86%)	0.288
Yes	1 (2%)	0		3 (6%)	2 (14%)	

EPE = extraprostatic extension; SVI = seminal vesicle invasion.

**Table 4 diagnostics-14-02751-t004:** Change in treatment strategies before and after PSMA PET/CT.

Patient *n*#	Risk	Initial Treatment	Post-PET Treatment	Change in Management
1	Favorable	Surgery	Systemic therapy	Major
2	Favorable	Radiotherapy	RT planning	Minor
3	Unfavorable	Radiotherapy	RT planning	Minor
4	Unfavorable	Radiotherapy	Systemic therapy	Major
5	Unfavorable	Radiotherapy	Systemic therapy	Major
6	Favorable	Radiotherapy	RT planning	Minor
7	Unfavorable	Radiotherapy	Systemic therapy	Major
8	Favorable	Surgery	Tailoring of LND template	Minor
9	Favorable	Surgery	Tailoring of LND template	Minor
10	Favorable	Surgery	Tailoring of LND template	Minor
11	Unfavorable	Radiotherapy	Systemic therapy	Major
12	Unfavorable	Radiotherapy	RT planning	Minor
13	Favorable	Surgery	Tailoring of LND template	Minor
14	Favorable	Radiotherapy	RT planning	Minor
15	Favorable	Surgery	Tailoring of LND template	Minor
16	Unfavorable	Radiotherapy	RT planning and systemic treatment	Major
17	Unfavorable	Surgery	Tailoring of LND template	Minor
18	Favorable	Radiotherapy	RT planning	Minor
19	Unfavorable	Radiotherapy	Systemic therapy	Major
20	Unfavorable	Radiotherapy	Systemic therapy	Major
21	Unfavorable	Radiotherapy	RT planning	Minor
22	Favorable	Surgery	Systemic therapy	Major
23	Unfavorable	Radiotherapy	RT planning	Minor
24	Unfavorable	Surgery	Systemic therapy	Major

## Data Availability

The original contributions presented in this study are included in the article/Appendix A. Further inquiries can be directed to the corresponding author.

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
