# Peer review of "The Evaluation of Radiolabeled Prostate-Specific Membrane Antigen Positron Emission Tomography/Computed Tomography for Initial Staging in Intermediate-Risk Prostate Cancer Patients: A Retrospective Multicenter Analysis"

_diagnostics, 2024, doi:10.3390/diagnostics14232751_

Round 1
Reviewer 1 Report
Comments and Suggestions for Authors
Well outlined introduction and rationale of the study: The authors aimed to assess the diagnostic performance of PSMA PET/CT in intermediate-risk prostate cancer. They also look at the performance of PSMA PET/CT in favourable vs unfavourable as an objective.
They elaborate on the role pitfalls of some of the validated methods such as CT with bone scan, some scores. They also emphasize that most studies have evaluated the peformance in mixed populations(Intermediate and high risk), with none of the studies to the best of their knowledge have looked at intermediate risk only. This is a relevant study.
Just two comments: The authors mentioned that the images were intepreted visually . They further mentioned the use of the primary score e.g Score 5: SUVmax >12. This is misleading.
The authors need to make it clear what the intended therapies were prior to the PSMA PET-CT. Is it possible to provide such information? Are the authors assuming that every patient needed localized therapy and whatever the MDT decides will be regarded as the change? or was there an MDT prior to requesting a PSMA PET/CT? Suggest expanding on Section 2.5: Change of management. This also includes the methods section in the abstract; alternatively, this can be provided as a supplementary table.
Well-structured manuscript.
Author Response
Well outlined introduction and rationale of the study: The authors aimed to assess the diagnostic performance of PSMA PET/CT in intermediate-risk prostate cancer. They also look at the performance of PSMA PET/CT in favourable vs. unfavourable as an objective.
They elaborate on the role pitfalls of some of the validated methods, such as CT with bone scans and some scores. They also emphasize that most studies have evaluated the performance in mixed populations (intermediate and high risk), with none of the studies, to the best of their knowledge, having looked at intermediate risk only. This is a relevant study.
Just two comments:
Q1. The authors mentioned that the images were interpreted visually. They further mentioned the use of the primary score, e.g., Score 5: SUVmax > 12. This is misleading.
R1. We are thankful to the reviewer for this comment. We have now included more details in the materials and methods section about the semiquantitative analysis, useful for the correct identification of the primary score.
Q2. The authors need to make it clear what the intended therapies were prior to the PSMA PET-CT. Is it possible to provide such information? Are the authors assuming that every patient needed localized therapy and whatever the MDT decides will be regarded as the change? or was there an MDT prior to requesting a PSMA PET/CT? Suggest expanding on Section 2.5: Change of management. This also includes the methods section in the abstract; alternatively, this can be provided as a supplementary table.
R2. We are thankful to the reviewer for the kind comment. Indeed, we have included more details in the specific subparagraph about the change of treatment strategy by including a new table (Table 4).
Well-structured manuscript.
Thanks a lot!
Reviewer 2 Report
Comments and Suggestions for Authors
The use of PET-PSMA studies in the initial staging of prostate cancer is becoming more established.
As reflected in the article, patient selection is a fundamental issue as it determines the cost-effectiveness of the test.
The results of the study point to a change in patient management in 13% of cases, mainly in intermediate risk patients in the unfavorable disease group.
However, we do not yet have results on how this change in management affects patient outcomes, a key point for establishing PET-PSMA as a key element in the staging of prostate cancer. To this end, it is necessary to include PET-PSMA as a diagnostic test alongside standard techniques in clinical trials.
Author Response
The use of PET-PSMA studies in the initial staging of prostate cancer is becoming more established.
Q3. As reflected in the article, patient selection is a fundamental issue as it determines the cost-effectiveness of the test. The results of the study point to a change in patient management in 13% of cases, mainly in intermediate risk patients in the unfavorable disease group. However, we do not yet have results on how this change in management affects patient outcomes, a key point for establishing PET-PSMA as a key element in the staging of prostate cancer. To this end, it is necessary to include PET-PSMA as a diagnostic test alongside standard techniques in clinical trials.
R3. We are thankful to the Reviewer for this comment. We have included a small sentence in the conclusion paragraph to reinforce the utility of PET-PSMA as a standard diagnostic technique in the future clinical trials.